# Incarceration status and cancer mortality: A population-based study

**Oluwadamilola T. Oladeru**[1]*, **Jenerius A. Aminawung**[2,3], **Hsiu-Ju Lin**[4,5], **Lou Gonsalves**[6], **Lisa Puglisi**[2], **Sophia Mun**[7], **Colleen Gallagher**[8], **Pamela Soulos**[3], **Cary P. Gross**[3], **Emily A. Wang**[2]

1 Department of Radiation Oncology, University of Florida, Gainesville, Florida, United States of America,
2 Department of Internal Medicine, SEICHE Center for Health and Justice, Yale School of Medicine, New Haven, Connecticut, United States of America, 3 Department of Internal Medicine, Yale Cancer Outcomes, Public Policy, and Effectiveness Research (COPPER) Center, New Haven, Connecticut, United States of America, 4 School of Social Work, University of Connecticut, Storrs, Connecticut, United States of America, 5 Research Division, Connecticut Department of Mental Health and Addiction Services, Hartford, Connecticut, United States of America, 6 Connecticut Tumor Registry, Connecticut Department of Public Health, Hartford, Connecticut, United States of America, 7 Kaiser Permanente Washington Health Research Institute, Seattle, Washington, United States of America, 8 Connecticut Department of Correction, Wethersfield, Connecticut, United States of America

* ooladeru@ufl.edu

**Data Availability Statement:** The data that support the findings of this study are available from the Connecticut Department of Public Health, but restrictions apply to the availability of these data,

## Abstract

### Background

The complex relationship between incarceration and cancer survival has not been thoroughly evaluated. We assessed whether cancer diagnosis during incarceration or the immediate post-release period is associated with higher rates of mortality compared with those never incarcerated.

### Methods

We conducted a population-based study using a statewide linkage of tumor registry and correctional system movement data for Connecticut adult residents diagnosed with invasive cancer from 2005 through 2016. The independent variable was place of cancer diagnosis: during incarceration, within 12 months post-release, and never incarcerated. The dependent variables were five-year cancer-related and overall survival rates.

### Results

Of the 216,540 adults diagnosed with invasive cancer during the study period, 239 (0.11%) people were diagnosed during incarceration, 479 (0.22%) within 12 months following release, and the remaining were never incarcerated. After accounting for demographics and cancer characteristics, including stage of diagnosis, the risk for cancer-related death at five years was significantly higher among those diagnosed while incarcerated (AHR = 1.39, 95% CI = 1.12–1.73) and those recently released (AHR = 1.82, 95% CI = 1.57–2.10) compared to the never-incarcerated group. The risk for all-cause mortality was also higher for those diagnosed with cancer while incarcerated (AHR = 1.92, 95% CI = 1.63–2.26) and those recently released (AHR = 2.18, 95% CI = 1.94–2.45).

which were used under Data Use Agreement for the current study, and so are not publicly available. Data are however available from the authors upon reasonable request and with permission of the Connecticut Department of Public Health.Anyone interesting in obtaining the dataset we used for our analysis will need to submit a complete application and obtain approval from the human investigations committee of the Connecticut department of public health (dph.hic@ct.gov).

**Funding:** This work is supported by the National Institutes of Health R01 5R01CA230444-02, awarded to CPG and EAW. The National Institutes of Health had no role in the design and conduct of the study; management, analysis, and interpretation of the data; manuscript preparation, and decision to submit the manuscript for publication.

**Competing interests:** I have read the journal's policy and the authors of this manuscript have the following competing interests. OTO reports funding unrelated to submitted work from Radiation Oncology Institute, NRG Oncology, Pfizer/ASCO Foundation and Bristol Meyers Squibb Foundation. CPG has received research funding NCCN Foundation (funds provided by AstraZeneca), Genentech as well as funding from Johnson and Johnson to help devise and implement new approaches sharing clinical trial data. The other authors have no competing interests to disclose. This does not alter our adherence to PLOS ONE policies on sharing data and materials.

## Conclusions and relevance

There is a higher risk of cancer mortality among individuals diagnosed with cancer during incarceration and in the first-year post-release, which is not fully explained by stage of diagnosis. Cancer prevention and treatment efforts should target people who experience incarceration and identify why incarceration is associated with worse outcomes.

## Introduction

The United States has the highest per capita rate of incarceration (698 per 100,000) in the world [1], and disproportionately incarcerates people of racial and ethnic minority backgrounds and those of low socioeconomic status [2]. Cancer is the leading cause of death among people incarcerated in prison, accounting for about 30% of all deaths [3]. Prior work has suggested that being incarcerated is associated with shorter survival after a cancer diagnosis. One single-center study found that the median survival was substantially inferior (21 months for incarcerated versus 54 months in non-incarcerated) [4], although this analysis did not account for cancer type or stage at diagnosis. Moreover, the impact of incarceration on health outcomes for people with cancer may persist even after incarceration. Medicare beneficiaries recently released from correctional facilities had higher cancer-related hospitalization rates compared with the general population in the months following release [5]. Other population-based studies have also found an increased risk of cancer mortality among incarcerated individuals compared with the general population following release [6–10]. However, none of these studies ascertained place of cancer diagnosis to distinguish cancers diagnosed during or after release from incarceration.

Critical knowledge gaps regarding the relation between incarceration, clinical factors, and cancer outcomes remain [11]. On the one hand, having been incarcerated may counterintuitively improve cancer outcomes for certain populations, given a constitutionally guaranteed access to healthcare in correctional facilities [12]. On the other hand, people who are incarcerated may face barriers to cancer screening or evaluation of symptoms–either prior to or during incarceration–and therefore present at a later stage. Indeed, recent evidence from a single urban tertiary care center shows that cancer is diagnosed at more advanced stages among incarcerated people [13]. After release from correctional facilities, access to community healthcare could be impaired due to barriers in obtaining public insurance, housing, and employment compounded by the lasting impact of a criminal record, leading to low rates of screening and barriers to cancer treatment [14].

In order to identify fundamental contributors to racial and ethnic cancer disparities, it is critical to further our understanding of the relationship between the carceral system and cancer outcomes. We therefore examined whether being diagnosed with cancer during incarceration or immediately post release is associated with an increased risk of cancer-related and all-cause mortality using a statewide data linkage. We also assessed whether stage at diagnosis might account for the relationship. We hypothesized that diagnosis of cancer while incarcerated or in the immediate post release period would be independently associated with a higher likelihood of cancer mortality than those never incarcerated, and this association would be independent of stage at diagnosis.

## Methods

### Study design and data sources

We created a statewide linkage of administrative data from the Connecticut Tumor Registry (CTR) and Connecticut Department of Correction (CT DOC) to assess the association between place of cancer diagnosis and mortality among patients diagnosed with invasive

cancer in Connecticut from 2005–2016. The CTR is a population-based registry that includes all reported cancers diagnosed in Connecticut residents since 1935, including data on the first course of treatment and follow-up data for estimating cancer survival. All medical providers, hospitals, and private pathology laboratories are required by state law to report new cancer diagnoses to the registry, including records on incarcerated persons. CT DOC is one of six unified correctional systems in the United States with all jails and prisons supervised by a single agency, making it easier to ascertain any admissions into the state's jails or prisons.

To identify people with a diagnosis of cancer during the study period and the place of diagnosis, we linked individuals with any incarceration history in the CT DOC master files between 2005 and 2016 to tumor registry data, from the same period, using name, date of birth, sex, race and ethnicity, and social security number. Individuals with cancer that did not have a match in the CT DOC file during the study period were considered the never-incarcerated group. For individuals in the CT DOC file that matched with the tumor registry, we used the CT DOC movement files, which has information on when individuals are admitted and released, to determine whether they were diagnosed while incarcerated or within 12 months of release. Information extracted from the CTR included age at cancer diagnosis, month and year of cancer diagnosis, cancer stage, type of cancer, vital status (deceased or alive), number of survival days from cancer diagnosis until the date of death or end of study period, and the primary cause of death. Details of our study protocol and partnership with relevant stakeholders of this study have been previously described in full detail [15]. The Yale University Institutional Review Board, the Connecticut Department of Public Health Human Investigations Committee and the CT DOC Research Advisory Council approved the study. Participant consent was waived by Yale University Institutional Review Board, and the Connecticut Department of Public Health Human Investigations Committee. This study follows the Strengthening the Reporting of Observational Studies in Epidemiology (STROBE) reporting guideline.

## Study sample

We included only cases with invasive cancers and selected the first diagnosed invasive cancer during the study period as the index cancer. If multiple invasive cancers were diagnosed synchronously, the more advanced stage case was designated the index cancer. If a person had more than one invasive cancer diagnosed on the same day and at the same stage, the cancer associated with primary cause of death was selected as the index cancer. We excluded: (1) individuals with sex defined as "transsexual" or "other" per the Department of Public Health data use requirements that prohibit reporting data of small cell sizes to protect confidentiality of individuals and minimize incorrect conclusions; (2) individuals younger than 18 years of age; (3) in-situ cancer diagnoses; and (4) people with index cancers diagnosed more than 12 months after release from a correctional facility, as the effect of incarceration may be more difficult to disentangle from other community factors.

## Measures

The independent variable in our study was place of cancer diagnosis, categorized as follows: 1) Individuals diagnosed during incarceration (incarcerated group); 2) individuals diagnosed within 12 months following release from incarceration (recently released group); and 3) individuals with no exposure to the criminal justice system during the study period (never incarcerated). The primary outcome measures were five -year cancer-related mortality, defined as death from cancer based on ICD-10 codes, and survival time.

Study covariates included participant demographic and index cancer characteristics. We created categories for age (18–30 years, >30 to 50, >50 to 70, and >70); race and ethnicity

(non-Hispanic Black, Hispanic, non-Hispanic white, and other); and sex (male or female). Race and ethnicity were derived from the CTR and were ascertained from documentation provided by the healthcare provider, system, or pathology lab which reported the cancer diagnosis. To eliminate groups with small sample sizes, we described cancer stage as 1) localized, including Stage I cancers localized to organ; 2) regional, including Stage II and III cancers that are larger and involve regional nodes; 3) distant which is defined by Stage IV cancers; or 4) unknown, defined as unstaged cancers. We grouped cancers with the potential of early detection through screening or surveillance (breast, colorectal, cervical, and prostate) as screenable and all others as non-screenable. We excluded lung and liver cancer from the screenable cancer group because there were no broadly-implemented screening guidelines applicable to these two cancers during our study period. We also grouped cancers by organ systems into the following categories to avoid small cell sizes: breast, gastrointestinal, thoracic, genitourinary (non-reproductive organs), male reproductive, female reproductive, leukemia/lymphomas, central nervous system, head and neck, sarcoma, skin, and others Table in **S1 Table**.

## Data analysis

We compared demographic and cancer characteristics by place of cancer diagnosis (in a correctional facility, recently released, or in the community and never incarcerated) using Chi-square or Fisher exact tests. To maintain the confidentiality of individual records, we did not report results that include fewer than five unweighted cases in any cell [16].

We used the Kaplan-Meier method to determine the five-year survival rate by place of diagnosis. Survival time in days was calculated from the date of index cancer diagnosis through the date of death. Patients who were alive five years after the index cancer diagnosis were censored at 1,825 days, and cases with follow-up time less than five years who did not die during follow-up were censored at the end of the study period (12/31/2018). To protect patient confidentiality, we only received the month and year of the cancer diagnosis from the CTR. We performed a sensitivity analysis using the first, fifteenth, and thirtieth day of the month as the date of cancer diagnosis, which did not yield any substantive differences. Herein, we report the outcome models based on diagnosis on the fifteenth of the month.

We then used Cox proportional hazard regression models to determine differential survival risk between those diagnosed during incarceration, following release, and in the community. The first model was performed for cancer-related death and a secondary model for death from all causes. Survival time for the Cox survival analysis was calculated in a fashion similar to the 5-year survival rate, but in lieu of 5-years of follow-up duration, all patients were censored at the end of the 2018 calendar year. Cox models included the primary predictor (place of diagnosis, with those never incarcerated as a reference group), as well as all other covariates, including age group, race and ethnicity, sex, and screenable versus non-screenable cancers. We also included stage of cancer diagnosis to examine its contribution to the risk of cancer-related mortality and all-cause mortality in this population. We considered an association significant for a p-value <0.05 and calculated adjusted hazards ratios (AHRs) with 95% confidence intervals. We conducted all analyses in SPSS version 26 [17].

## Results

Overall, 216,540 individuals were diagnosed with an invasive cancer in Connecticut during the study period; 239 of whom were diagnosed in prison, 479 within 12 months post-release, and 215,822 with no exposure to incarceration. The mean length of incarceration for those diagnosed while incarcerated was 4.6 years (SD = 6.1), and 1.1 years (SD = 1.1) for the recently released group. On average individuals had a cancer diagnosis 3.6 years (SD = 5.7) following

admission into prison, and 5.1 months (SD = 3.5) after release Table in **S2 Table**. The median age at diagnosis was 50 years for those incarcerated, 51 years for those within 12 months post-release and 66 years for those never incarcerated. In the incarcerated group 72.4% had been incarcerated for a year or longer, and 26.1% in recently released group were incarcerated for at least 1 year.

People diagnosed with invasive cancer while incarcerated and within 12 months post-release were more likely to be younger, male, and non-Hispanic Black or Hispanic compared with those never incarcerated (p<0.001) (**Table 1**). Cancers originating from the gastrointestinal system (dominated by colorectal and liver) were the most common cancers in all three groups (incarcerated, 25.5%; recently released, 32.8%; and never incarcerated, 17.8%, respectively). Other cancers commonly diagnosed among those incarcerated included those in the thoracic region (97% of which were lung), the male reproductive organs (62% were prostate), and leukemia and lymphomas. Incarcerated individuals were diagnosed with cancer at a distant stage more frequently compared to those recently released or never incarcerated (42.7% vs. 28.4% vs 25.0%, p<0.001) (**Table 1**). The average length of incarceration for individuals diagnosed with screenable cancers at a distant stage (metastatic) in the incarcerated group was 5.9 (SD = 7.5) years, compared with 4.8 (SD = 5.9) years for those with localized screenable cancers (p = 0.58). For screenable cancers, 58.8% were diagnosed at a distant stage among incarcerated group; 40.8% for recently released group, and only 31.9% in the never incarcerated group.

The 5-year survival rate was lowest among individuals diagnosed within 12 months of release from a correctional facility (54.6%; 95% Confidence Interval (CI): 46.8% - 62.4%), compared to 63.2% (95%CI: 55.4% - 71.0%) among those diagnosed while incarcerated, and 67.2% (95%CI: 67.0% - 67.4%) among the never incarcerated group (**Table 2**). For the subset of patients with screenable cancers, the 5-year survival rate was 67.4% (95%CI: 53.4% - 81.5%) for individuals diagnosed while incarcerated, 77.6% (95%CI: 69.5% - 85.6%) for individuals diagnosed within 12-months post-release and 85.2% (95%CI: 84.9% - 85.5%) among those never incarcerated. Five-year survival rates for specific cancers also differed by place of diagnosis. For breast cancer, the 5-year survival rate was lowest for incarcerated patients (60%), compared to those within 12-months after release (81.8%) and those never incarcerated (89.5%); a pattern similar to that observed for all screenable cancers given the small number of women and breast cancers diagnosed in the incarcerated group.

The cumulative survival for all causes of death was similar for both groups with criminal justice exposure and significantly lower compared with the never incarcerated group (**Fig 1A**). In adjusted analyses, individuals diagnosed while incarcerated had worse all-cause mortality (Adjusted Hazard Ratio (AHR) = 1.92, 95% CI, 1.63–2.26) compared with those never incarcerated, as did those diagnosed immediately post release (AHR = 2.18, 95% CI 1.94–2.45) (**Table 3**)

Cancer-specific survival was also lowest among the group diagnosed after release, followed by those diagnosed while incarcerated, and highest amongst those never incarcerated (**Fig 1B**). Individuals diagnosed while incarcerated had worse cancer-related mortality (AHR) = 1.39, 95% CI, 1.12–1.73) compared with those never incarcerated. Those diagnosed immediately post release had a higher risk for cancer-related mortality (AHR = 1.82, 95% CI 1.57–2.10) compared to those never incarcerated (**Table 4**).

Stage of diagnosis accounted for some of the relation between incarceration status and mortality but was not the sole mediator. For instance, the risk of cancer-related death among patients who were diagnosed while incarcerated changed from 1.73 (95%CI: 1.40–2.15) to 1.39 (95%CI: 1.12–1.73) after adjusting for stage at diagnosis. Among patients diagnosed within 12 months of release, the hazard ratio changed from 1.91 (95%CI: 1.65–2.21) to 1.82 (95%CI: 1.57–2.10) after adjusting for stage at diagnosis (**Table 4**).

**Table 1. Characteristics of individuals diagnosed with invasive cancer between 2005 and 2016 cancer by incarceration status at cancer diagnosis.**

| Characteristic | Incarcerated (N = 239) | 12-months post release (N = 479) | Never incarcerated (N = 215,822) | p-value |
|---|---|---|---|---|
| | No. (%) | No. (%) | No. (%) | |
| **Age at diagnosis (years)** | | | | <0.001 |
| 18–30 | 22 (9.2) | 25 (5.2) | 3489 (1.6) | |
| >30–50 | 102 (42.7) | 197 (41.1) | 28389 (13.2) | |
| >50–70 | 104 (43.5) | 241 (50.3) | 99237 (46.0) | |
| >70 | 11 4.6) | 16 (3.3) | 84707 (39.2) | |
| **Race and Ethnicity*** | | | | <0.001 |
| Non-Hispanic White | 112 (46.9) | 236 (49.3) | 181358 (84.0) | |
| Non-Hispanic Black | 83 34.7) | 156 (32.6) | 14913 (6.9) | |
| Hispanic | >39 (>16.3) | >82 (>17.1) | 13677 (6.3) | |
| Other | <5 (<2.1) | <5* (<1.0) | 5874 (2.7) | |
| **Sex** | | | | <0.001 |
| Male | 216 (90.4) | 412 (86.0) | 103317 (47.9) | |
| Female | 23 (9.6) | 67 (14.0) | 112505 (52.1) | |
| **Cancer Type (categories)*** | | | | <0.001 |
| Breast | 6 (2.5) | 25 (5.2) | 34622 (16.0) | |
| Gastrointestinal | 61 (25.5) | 157 (32.8) | 38376 (17.8) | |
| Thoracic | 38 (15.9) | 64 (13.4) | 28247 (13.1) | |
| Male Reproductive | 29 (12.1) | 68 (14.2) | 31120 (14.4) | |
| Leukemia & Lymphoma | 37 (15.5) | 45 (9.4) | 19743 (9.1) | |
| Urinary (non-reproductive) | 11 (4.6) | 37 (7.7) | 12683 (5.9) | |
| Skin | 5 (2.1) | 6 (1.3) | 10268 (4.8) | |
| Head and Neck | 28 (11.7) | 40 (8.4) | 14856 (6.9) | |
| Female Reproductive | 5 (2.1) | 6 (1.3) | 13299 (6.2) | |
| Central Nervous System (CNS) | <5 (<2.1) | 8 (1.7) | 3123 (1.4) | |
| Sarcoma | 8 (3.3) | 12 (2.5) | 1823 (0.8) | |
| Other | >6 (>2.5) | 11 (2.3) | 7662 (3.6) | |
| **Cancer stage** | | | | <0.001 |
| Localized | 72 (30.1) | 185 (38.6) | 103561 (48.0) | |
| Regional | 56 (23.4) | 130 (27.1) | 45715 (21.2) | |
| Distant | 102 (42.7) | 136 (28.4) | 53971 (25.0) | |
| Unknown/unstaged | 9 (3.8) | 28 (5.8) | 12575 (5.8) | |
| **Screenable cancer**** | | | | <0.001 |
| Screenable | 51 (21.3) | 120 (25.1) | 84551 (39.2) | |
| Non-screenable | 188 (78.7) | 359 (74.9) | 131271 (60.8) | |

Note: No = Number

* cell sizes containing fewer than 5 individuals are suppressed due to privacy concerns

**screenable cancer types include breast, colorectal, cervical, and prostate.

## Discussion

Using data from a statewide population-based linkage, we found that being diagnosed with cancer while incarcerated or within 12 months following release was associated with higher rates of cancer-related and all-cause mortality compared with people never incarcerated, even after controlling for participant demographics, type of cancer, and stage of cancer at diagnosis. Possible reasons for the high risk of death include having limited access to high quality cancer care or even experimental cancer treatment, such as participation in clinical trials, access to

**Table 2. Five-year survival rates by cancer type and by incarceration status at cancer diagnosis.**

| Cancer type | Incarcerated | 12-months post release | Never incarcerated |
|---|---|---|---|
| | Rate% (95%CI) | Rate% (95%CI) | Rate% (95%CI) |
| Breast | 60.0 (16.9–100.0) | 81.8 (64.2–99.4) | 89.5 (89.5–89.5) |
| Gastrointestinal | 43.2 (27.5–58.9) | 27.8 (18.0–37.6) | 47.5 (47.5–47.5) |
| Thoracic | 42.3 (22.7–61.9) | 24.1 (10.4–37.8) | 28.5 (28.5–28.5) |
| Male Reproductive | 91.8 (80.0–100.0) | 100.0 (100.0–100.0) | 92.2 (92.2–92.2) |
| Leukemia & Lymphoma | 66.0 (46.4–85.6) | 75.5 (57.9–93.1) | 68.6 (68.6–68.6) |
| Genitourinary (non-reproductive) | 62.5 (29.2–95.8) | 70.0 (50.4–89.6) | 67.0 (67.0–67.0) |
| Skin | 100.0 (100.0–100.0) | 100.0 (100.0–100.0) | 88.5 (88.5–88.5) |
| Head and Neck | 72.1 (52.5–91.7) | 76.5 (60.8–92.2) | 82.9 (82.9–82.9) |
| Female Reproductive | 100.0 (100.0–100.0) | 50.0 (10.8–89.2) | 71.3 (71.3–71.3) |
| Central Nervous System | 0 (0.0–0.0) | 66.7 (13.8–100.0) | 38.8 (36.8–40.8) |
| Sarcoma | 66.7 (29.5–100.0) | 81.8 (50.4–100.0) | 66.9 (64.9–68.9) |
| Other | 73.3 (41.9–100.0) | 42.9 (5.7–80.1) | 54.9 (52.9–56.9) |
| Screenable cancers* | 67.4 (53.4–81.5) | 77.6 (69.5–85.6) | 85.2 (84.9–85.5) |
| Non Screenable cancers | 56.8 (49.0–64.6) | 49.8 (12.6–87.1) | 55.7 (53.7–57.6) |
| Total | 63.2 (55.4–71.0) | 54.6 (46.8–62.4) | 67.2 (67.0–67.4) |

Note: 95%CI = 95 percent confidence interval

* screenable cancer types include breast, colorectal, cervical, and prostate.

palliative care, and attention to patient's social determinants of health, including social support and food [18–22].

While prior studies have identified the association between incarceration and cancer survival time [13], our study illuminates the immediate post release period as a particularly high-risk time period. People diagnosed shortly following release from correctional settings are more likely to die earlier from cancer compared with counterparts diagnosed during incarceration, likely reflecting the significant barriers following release for obtaining timely cancer care. People are often released from correctional facilities without health insurance, medical records, or a primary care appointment, while also contending with severe structural barriers to obtaining housing, food, and employment [23, 24]. While Medicaid coverage expanded in Connecticut during this study, recent work has shown that insurance is necessary but not sufficient to engage people released from correctional settings into mental health or substance use treatment [25, 26]. This may also be the case for cancer treatment. Primary care for people recently released from correctional systems should include screening for treatable cancers, evaluation of symptoms, and addressing social determinants to mitigate these disparities in cancer related deaths.

Our findings regarding stage of cancer diagnosis and setting of cancer diagnosis are striking. Almost two-thirds of patients diagnosed with cancer while incarcerated and more than half of those diagnosed following release were found to have regional spread or metastasized cancer, including for screenable cancers such as colorectal cancer. These data support the previously reported trend of late-stage cancer diagnosis in the incarcerated population [13]. Past research has indicated that incarcerated individuals are less likely to receive age-appropriate screening and this disparity warrants immediate health system intervention [27].

Our study also corroborates prior findings of distant-stage diagnosis of lung cancer and colorectal cancer among the incarcerated population. Despite the increased risk for lung cancer in the justice-involved population, the routine use of low-dose computed tomography

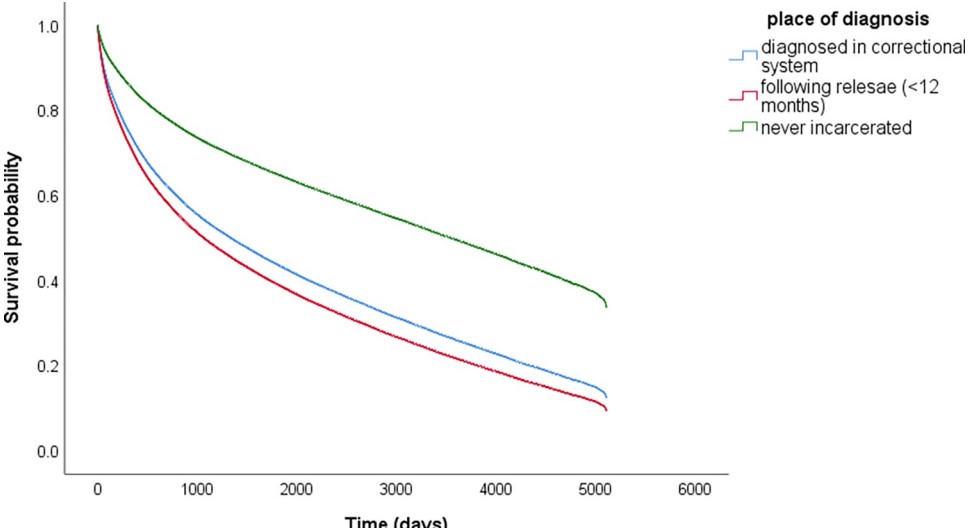

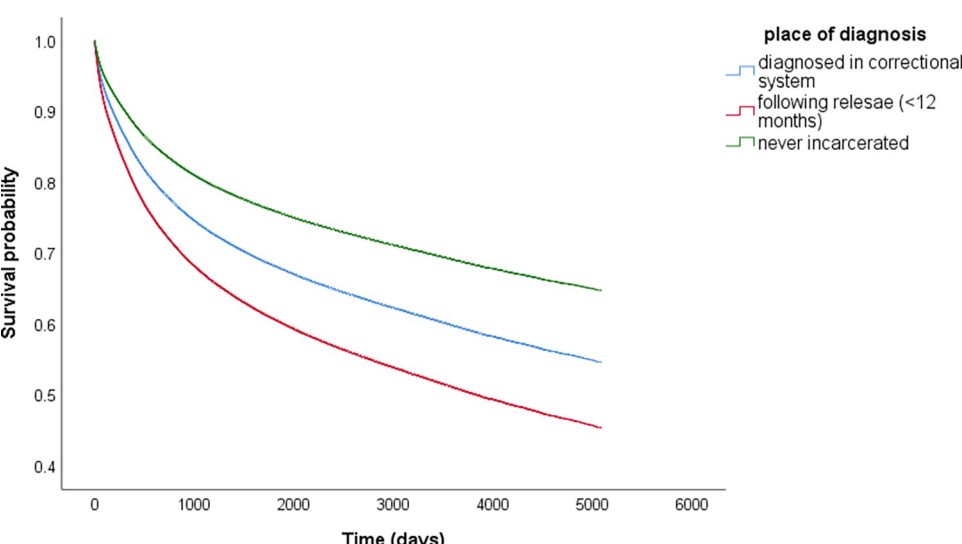

**Fig 1. a**: Kaplan Meier curves of all-cause mortality by place of diagnosis. Green = Never incarcerated; Blue = Incarcerated; Red = Post-release **b**: Kaplan Meier curves of Cancer-related mortality by place of diagnosis. Green = Never incarcerated; Blue = Incarcerated; Red = Post-release.

scans for screening in individuals with a significant smoking history, as recommended by the US Preventive Services Task Force, has yet to be adopted in correctional facilities [28]. Colonoscopies are also not widely available [27]. Further, past studies have documented that justice involved men may be hesitant to get screened, even when colonoscopies are available, due to fatalistic thinking about cancer [29]. While there is no systematic study examining screening rates among incarcerated people or people who have been incarcerated, past single site studies have shown low rates of preventive cancer screening overall [30, 31]. Given the rising incidence of colorectal cancers among people younger than 50, especially among young Black adults, targeting this population for universal access to screening and health information

**Table 3. Hazard ratios for all-cause mortality.**

| | Model not adjusted for cancer stage | | Model adjusted for cancer stage | |
|---|---|---|---|---|
| | AHR (95%CI) | p-value | AHR (95%CI) | p-value |
| **Status at diagnosis** | | | | |
| Never incarcerated | 1.00 (ref) | | 1.00 (ref) | |
| Incarcerated | 2.31 (1.96–2.72) | <0.001 | 1.92 (1.63–2.26) | <0.001 |
| Post-release (≤12 months) | 2.30 (2.05–2.59) | <0.001 | 2.18 (1.94–2.45) | <0.001 |
| **Age** | | | | |
| 18–30 years | 1.00 (ref) | | 1.00 (ref) | |
| >30–50 years | 2.17 (1.97–2.40) | <0.001 | 2.12 (1.93–2.34) | <0.001 |
| >50–70 year | 4.68 (4.26–5.15) | <0.001 | 4.32 (3.93–4.75) | <0.001 |
| >70 years | 12.60 (11.47–13.85) | <0.001 | 10.93 (9.95–12.02) | <0.001 |
| **Race and ethnicity** | | | | |
| Non-Hispanic White | 1.00 (ref) | | 1.00 (ref) | |
| Non-Hispanic Black | 1.28 (1.25–1.31) | <0.001 | 1.19 (1.16–1.22) | <0.001 |
| Hispanic | 1.03 (1.01–1.06) | 0.02 | 0.97 (0.94–0.99) | <0.001 |
| Other | 0.57 (0.54–0.60) | <0.001 | 0.53 (0.50–0.56) | <0.001 |
| **Gender** | | | | |
| Male | 1.00 (ref) | | 1.00 (ref) | |
| Female | 0.89 (0.88–0.90) | <0.001 | 0.89 (0.88–0.90) | <0.001 |
| **Cancer type** | | | | |
| Not screenable cancer | 1.00 (ref) | | 1.00 (ref) | |
| Screenable* cancer | 0.38 (0.37–0.38) | <0.001 | 0.57 (0.56–0.58) | <0.001 |
| **Cancer stage** | | | | |
| Localized | — | — | 1.00 (ref) | |
| Regional | — | — | 2.10 (2.06–2.14) | <0.001 |
| Distant | — | — | 4.25 (4.18–4.32) | <0.001 |
| Unknown/unstaged | — | — | 4.94 (4.82–5.05) | <0.001 |

Note: AHR = Adjusted hazard ratio; 95%CI = 95 percent confidence interval

* screenable cancer types include breast, colorectal, cervical, and prostate.

about cancer prevention through innovative digital platforms, may be important to reduce cancer disparities [32]. In addition, because the majority of those diagnosed with cancer while incarcerated had been incarcerated for more than a year, targeted cancer screening for those incarcerated for more than a year should be implemented. Our findings may be pertinent to other state prison systems as the national data reveal that cancer is now the leading cause of death among incarcerated individuals [3].

## Limitations

The small sample size for individuals diagnosed with cancer while incarcerated precluded examination of how cancer-specific mortality rates differed by place of diagnosis. However, the number of deaths in incarcerated individuals in our study are comparable to Bureau of Justice Statistics data on cancer deaths in correctional facilities in Connecticut [3]. Our measure of individuals' race and ethnicity and biologic sex were derived from medical records or health care providers and systems as provided to CTR, which could be less accurate than self-report. In addition, given the lack of established correctional guidelines on lung and liver cancer screening during our study period, this cancer type was excluded from our subset analysis of screenable cancers.

**Table 4. Hazard ratios for cancer-related mortality.**

| | Model not adjusted for cancer stage | | Model adjusted for cancer stage | |
|---|---|---|---|---|
| | AHR (95%CI) | p-value | AHR (95%CI) | p-value |
| **Status at diagnosis** | | | | |
| Never incarcerated | 1.00 (ref) | | 1.00 (ref) | |
| Incarcerated | 1.73 (1.40–2.15) | <0.001 | 1.39 (1.12–1.73) | 0.003 |
| Post-release (≤12 months) | 1.91 (1.65–2.21) | <0.001 | 1.82 (1.57–2.10) | <0.001 |
| **Age** | | | | |
| 18–30 years | 1.00 (ref) | | 1.00 (ref) | |
| >30–50 years | 2.37 (2.12–2.66) | <0.001 | 2.31 (2.06–2.58) | <0.001 |
| >50–70 year | 4.64 (4.15–5.18) | <0.001 | 4.14 (3.71–4.63) | <0.001 |
| >70 years | 9.75 (8.73–10.89) | <0.001 | 8.00 (7.16–8.93) | <0.001 |
| **Race and ethnicity** | | | | |
| Non-Hispanic White | 1.00 (ref) | | 1.00 (ref) | |
| Non-Hispanic Black | 1.26 (1.22–1.30) | <0.001 | 1.15 (1.12–1.18) | <0.001 |
| Hispanic | 1.00 (0.96–1.03) | 0.84 | 0.92 (0.89–0.95) | <0.001 |
| Other | 0.56 (0.52–0.60) | <0.001 | 0.52 (0.49–0.56) | <0.001 |
| **Gender** | | | | |
| Male | 1.00 (ref) | | 1.00 (ref) | |
| Female | 0.93 (0.91–0.94) | <0.001 | 0.93 (0.92–0.94) | <0.001 |
| **Cancer type** | | | | |
| Not screenable cancer | 1.00 (ref) | | 1.00 (ref) | |
| Screenable* cancer | 0.29 (0.28–0.29) | <0.001 | 0.49 (0.48–0.50) | <0.001 |
| **Cancer stage** | | | | |
| Localized | — | — | 1.00 (ref) | |
| Regional | — | — | 3.06 (2.99–3.13) | <0.001 |
| Distant | — | — | 6.70 (6.55–6.84) | <0.001 |
| Unknown/unstaged | — | — | 7.53 (7.31–7.76) | <0.001 |

Note: AHR = Adjusted hazard ratio; 95%CI = 95 percent confidence interval

* screenable cancer types include breast, colorectal, cervical, and prostate.

## Conclusion

Our study reveals that incarceration, whether at the time of receiving a cancer diagnosis or within the 12 months of release, is a significant determinant of disparities in cancer mortality. Interventions that improve preventive screening and quality cancer care for justice-involved populations are imperative components to attaining cancer health equity.

## Supporting information

**S1 Table. List of cancer ICD codes and grouping by organ system.**
(PDF)

**S2 Table. Length of incarceration and average time to cancer diagnosis.**
(PDF)

## Acknowledgments

Disclaimer

The Connecticut Department of Public Health does not endorse or assume any responsibility for any analyses, interpretations or conclusions based on the data. The authors assume full responsibility for all such analyses, interpretations, and conclusions.

## Author Contributions

**Conceptualization:** Jenerius A. Aminawung, Cary P. Gross, Emily A. Wang.

**Data curation:** Hsiu-Ju Lin.

**Formal analysis:** Hsiu-Ju Lin.

**Funding acquisition:** Cary P. Gross, Emily A. Wang.

**Investigation:** Cary P. Gross, Emily A. Wang.

**Methodology:** Oluwadamilola T. Oladeru, Jenerius A. Aminawung, Lisa Puglisi, Cary P. Gross, Emily A. Wang.

**Project administration:** Sophia Mun.

**Resources:** Lou Gonsalves, Colleen Gallagher.

**Supervision:** Cary P. Gross, Emily A. Wang.

**Validation:** Emily A. Wang.

**Writing – original draft:** Oluwadamilola T. Oladeru, Jenerius A. Aminawung, Emily A. Wang.

**Writing – review & editing:** Oluwadamilola T. Oladeru, Jenerius A. Aminawung, Hsiu-Ju Lin, Lou Gonsalves, Lisa Puglisi, Sophia Mun, Colleen Gallagher, Pamela Soulos, Cary P. Gross, Emily A. Wang.

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
