## [Decision Letter · Decision Letter 0]

4 May 2022

PONE-D-22-07747Incarceration status and cancer mortality: A population-based studyPLOS ONE

Dear Dr. Aminawung,

Thank you for submitting your manuscript to PLOS ONE. After careful consideration, we feel that it has merit but does not fully meet PLOS ONE’s publication criteria as it currently stands. Therefore, we invite you to submit a revised version of the manuscript that addresses the points raised during the review process. The reviewers and I were enthusiastic about the manuscript, although you will see some recommendations both in terms of methods (requesting additional description and additional consideration) and also framing (requesting inclusion of additional pieces to the introduction and discussion sections). I look forward to seeing the revised version.

We look forward to receiving your revised manuscript.

Kind regards,

Andrea Knittel

Academic Editor

PLOS ONE

Journal Requirements:

"This work is supported by the National Institutes of Health grant. Data on incident breast cancer cases used in this study were obtained from the Connecticut Tumor Registry located in the Connecticut Department of Public Health (DPH). Data on incarceration status were obtained from the Connecticut Department of Correction (DOC) master file of all individuals that interacted with the DOC during the study period and DOC movement files. The National Institutes of Health had no role in the design and conduct of the study; management, analysis, and interpretation of the data; manuscript preparation, and decision to submit the manuscript for publication.  The analysis, interpretation or conclusions drawn from these data are the responsibility of the authors and do not represent the views of neither the National Institutes of Health nor the United States Department of Health and Human Services or any of its affiliates."

We note that you have provided funding information. However, funding information should not appear in the Acknowledgments section or other areas of your manuscript. We will only publish funding information present in the Funding Statement section of the online submission form. 

"This work is supported by the National Institutes of Health R01 5R01CA230444-02, awarded to CPG and EAW. The National Institutes of Health had no role in the design and conduct of the study; management, analysis, and interpretation of the data; manuscript preparation, and decision to submit the manuscript for publication."

"I have read the journal's policy and the authors of this manuscript have the following competing interests. OTO reports funding unrelated to submitted work from Radiation Oncology Institute, NRG Oncology and Bristol Meyers Squibb Foundation. CPG has received research funding NCCN Foundation (funds provided by AstraZeneca), Genentech as well as funding from Johnson and Johnson to help devise and implement new approaches sharing clinical trial data. The other authors have no competing interests to disclose."

Reviewers' comments:

Reviewer's Responses to Questions

**Comments to the Author**

1. Is the manuscript technically sound, and do the data support the conclusions?

Reviewer #1: Yes

Reviewer #2: Yes

2. Has the statistical analysis been performed appropriately and rigorously? 

Reviewer #1: Yes

Reviewer #2: Yes

3. Have the authors made all data underlying the findings in their manuscript fully available?

Reviewer #1: No

Reviewer #2: No

4. Is the manuscript presented in an intelligible fashion and written in standard English?

Reviewer #1: Yes

Reviewer #2: Yes

5. Review Comments to the Author

Reviewer #1: Plos One

Important topic – that cancer is the leading cause of death among older prisoners – topic warrants investigation.

Intro – is it worth quantifying the cancer disparities faced by people with CJ involvement compared to the gen pop? Just to really drive home the point about significance?

Method – I realize it is not standard to report how the team got access to these data, but it might be useful to the field.

Otherwise, methods are clear.

Results – the gastro cancer findings are fascinating. Who knew so pervasive? Important finding.

Finding that men more likely to be diagnosed also interesting.

Discussion – how come no mention of women’s cancers? Is it not interesting that people diagnosed during incarceration more likely to be male? What’s that about?

Worth saying how and to what extent findings from CT can be extrapolated to other places?

Reviewer #2: Please see full comments in attachment.

6. PLOS authors have the option to publish the peer review history of their article (what does this mean?). If published, this will include your full peer review and any attached files.

Reviewer #1: No

Reviewer #2: **Yes: **Christopher Manz

---

## [Author Response · Author response to Decision Letter 0]

3 Aug 2022

Response to Reviewers Comments

Journal Requirements

Comment 1. Please ensure that your manuscript meets PLOS ONE's style requirements, including those for file naming. The PLOS ONE style templates can be found at 

Response: Thank you for the reminder. We have ensured this.

Comment 2. Please provide additional details regarding participant consent. In the ethics statement in the Methods and online submission information, please ensure that you have specified (1) whether consent was informed and (2) what type you obtained (for instance, written or verbal, and if verbal, how it was documented and witnessed). If your study included minors, state whether you obtained consent from parents or guardians. If the need for consent was waived by the ethics committee, please include this information.

Response: We have edited the manuscript study design section to include the following statement, “participant consent was waived by Yale University Institutional Review Board, and the Connecticut Department of Public Health Human Investigations Committee” 

Location: Page 6, line 117-119

Response: We have edited the manuscript study design section to include the following statement, “participant consent was waived by Yale University Institutional Review Board, and the Connecticut Department of Public Health Human Investigations committee. The limited dataset accessible to us included protected health information.” 

Location: Page 6, line 117-119

Comment 3. Thank you for stating the following in the Acknowledgments Section of your manuscript: 

"This work is supported by the National Institutes of Health grant. Data on incident breast cancer cases used in this study were obtained from the Connecticut Tumor Registry located in the Connecticut Department of Public Health (DPH). Data on incarceration status were obtained from the Connecticut Department of Correction (DOC) master file of all individuals that interacted with the DOC during the study period and DOC movement files. The National Institutes of Health had no role in the design and conduct of the study; management, analysis, and interpretation of the data; manuscript preparation, and decision to submit the manuscript for publication. The analysis, interpretation or conclusions drawn from these data are the responsibility of the authors and do not represent the views of neither the National Institutes of Health nor the United States Department of Health and Human Services or any of its affiliates."

We note that you have provided funding information. However, funding information should not appear in the Acknowledgments section or other areas of your manuscript. We will only publish funding information present in the Funding Statement section of the online submission form. 

"This work is supported by the National Institutes of Health R01 5R01CA230444-02, awarded to CPG and EAW. The National Institutes of Health had no role in the design and conduct of the study; management, analysis, and interpretation of the data; manuscript preparation, and decision to submit the manuscript for publication."

Response: We have removed the funding statement and ensured that the funding statement in the online submission is accurate and have included it in the cover letter. 

Location: Page 19, lines 341- 351 and updated cover letter 

Comment 4. Thank you for stating the following in the Competing Interests section: 

"I have read the journal's policy and the authors of this manuscript have the following competing interests. OTO reports funding unrelated to submitted work from Radiation Oncology Institute, NRG Oncology and Bristol Meyers Squibb Foundation. CPG has received research funding NCCN Foundation (funds provided by AstraZeneca), Genentech as well as funding from Johnson and Johnson to help devise and implement new approaches sharing clinical trial data. The other authors have no competing interests to disclose."

Response: We have added the requested statement in the form and cover letter.

Location: Online forms and cover letter.

Comment 5. In your Data Availability statement, you have not specified where the minimal data set underlying the results described in your manuscript can be found. PLOS defines a study's minimal data set as the underlying data used to reach the conclusions drawn in the manuscript and any additional data required to replicate the reported study findings in their entirety. All PLOS journals require that the minimal data set be made fully available. For more information about our data policy, please see http://journals.plos.org/plosone/s/data-availability.

Response: Legal restrictions to sharing data publicly has been emphasized in the edited cover letter 

Location: Cover letter

Response: We have addressed this request in our cover letter

Important: If there are ethical or legal restrictions to sharing your data publicly, please explain these restrictions in detail. Please see our guidelines for more information on what we consider unacceptable restrictions to publicly sharing data:http://journals.plos.org/plosone/s/data-availability#loc-unacceptable-data-access-restrictions. Note that it is not acceptable for the authors to be the sole named individuals responsible for ensuring data access.

Response: Thank you for the reminder. 

Location: Cover letter

Comment 6. We note that you have included the phrase “data not shown” in your manuscript. Unfortunately, this does not meet our data sharing requirements. PLOS does not permit references to inaccessible data. We require that authors provide all relevant data within the paper, Supporting Information files, or in an acceptable, public repository. Please add a citation to support this phrase or upload the data that corresponds with these findings to a stable repository (such as Figshare or Dryad) and provide and URLs, DOIs, or accession numbers that may be used to access these data. Or, if the data are not a core part of the research being presented in your study, we ask that you remove the phrase that refers to these data.

Response: We have removed such phrases “data not shown” from manuscript 

Location: Page 10, line 208

Comment 7. Please include captions for your Supporting Information files at the end of your manuscript, and update any in-text citations to match accordingly. Please see our Supporting Information guidelines for more information:http://journals.plos.org/plosone/s/supporting-information.

Response: N/A

Comment 8. Please review your reference list to ensure that it is complete and correct. If you have cited papers that have been retracted, please include the rationale for doing so in the manuscript text, or remove these references and replace them with relevant current references. Any changes to the reference list should be mentioned in the rebuttal letter that accompanies your revised manuscript. If you need to cite a retracted article, indicate the article’s retracted status in the References list and also include a citation and full reference for the retraction notice.

Response: We have reviewed the reference list and it is complete and free of retracted articles.

Location: Reference list.

Reviewers' comments:

Reviewer's Responses to Questions - Comments to the Author

Comment 1. Is the manuscript technically sound, and do the data support the conclusions?

Reviewer #1: Yes

Reviewer #2: Yes

Response: Thank you for your review.

Comment 2. Has the statistical analysis been performed appropriately and rigorously?

Reviewer #1: Yes

Reviewer #2: Yes

Response: Thank you for your comments.

Comment 3. Have the authors made all data underlying the findings in their manuscript fully available?

Reviewer #1: No

Reviewer #2: No

Response: We are unable to fully provide the underlying data of our findings as incarcerated individuals are considered vulnerable research population. Thus, participant privacy and legal restrictions prohibit us from sharing the underlying data for this study. 

Comment 4. Is the manuscript presented in an intelligible fashion and written in standard English?

Reviewer #1: Yes

Reviewer #2: Yes

Response: Thank you for the feedback.

5. Review Comments to the Author – Reviewer 1

Comment: Important topic – that cancer is the leading cause of death among older prisoners – topic warrants investigation.

Intro – is it worth quantifying the cancer disparities faced by people with CJ involvement compared to the gen pop? Just to really drive home the point about significance?

Response: We thank the reviewer for this feedback regarding our in-depth introduction section highlighting cancer disparities faced by people with criminal justice involvement. We felt it was important to use this medium as an opportunity to educate readers about an overlooked population in oncology, that is not only growing in size but also the notable change in leading cause of mortality previously known to be cardiovascular disease is now cancer. Furthermore, it provides an in-depth background on the various challenges of cancer care for this population, and we hoped to highlight the critical knowledge gap regarding incarceration and cancer outcomes. 

Location: No changes made.

Comment: Method – I realize it is not standard to report how the team got access to these data, but it might be useful to the field.

Otherwise, methods are clear.

Response: Thank you for this point; we agree that it is important for others in the field to know how the database was constructed. We have published our methodology for this project in full detail so that others can adopt our approach to accessing data and building a linkage registry for a population that is understandably difficult to investigate. Please see, Puglisi L, Halberstam AA, Aminawung J, Gallagher C, Gonsalves L, Schulman-Green D, et al. Incarceration and Cancer-Related Outcomes (ICRO) study protocol: using a mixed-methods approach to investigate the role of incarceration on cancer incidence, mortality and quality of care. BMJ open. 2021;11(5):e048863. In our methods section, we summarized how we created a statewide linkage registry using data from a tumor registry and department of corrections. We also described how we identified and linked individuals using pertinent identifying factors, matched movement files and selected for those with invasive cancer diagnosis. 

Location: “Details of our study protocol and partnership with relevant stakeholders of this study have been previously described in full detail (15).” Page 6, Lines 114-115 

Comment: Results – the gastro cancer findings are fascinating. Who knew so pervasive? Important finding. Finding that men more likely to be diagnosed also interesting.

Response: We were also surprised by this finding, and it further highlights the fact that prison health is public health. The rates of gastrointestinal cancers are increasing in the United States and now more pervasive in younger population. Thus, our findings highlight that our prison walls are not impermeable to similar trends we observe in the community. 

Location: No changes made.

Comment: Discussion – how come no mention of women’s cancers? Is it not interesting that people diagnosed during incarceration more likely to be male? What’s that about?

Response: Our data sample is heavily weighted towards men due to the characteristics of the incarcerated population. Yet, it is important to highlight the cancer outcomes for women to the extent that our data allow, given that there has been too little focus on the health of women who are incarcerated. In our results (and tables), we report the incidence and five-year survival rates of women with breast and female reproductive cancers. We included cervical and breast cancer in our subgroup analysis of screenable cancers. We had to group them together given the small number of individual cases, so as to limit the likelihood of unintentionally revealing the identity of participants. In our results, we have now included this sentence to highlight the unique 5-year survival rate from breast cancer between the three cohorts: “For breast cancer, the 5-year survival rate was lowest for incarcerated patients (60%), compared to those within 12-months after release (81.8%) and those never incarcerated (89.5%); a pattern similar to that observed for all screenable cancers given the small number of women and breast cancers diagnosed in the incarcerated group.” 

Location: Page 12, lines 226-229

Comment: Worth saying how and to what extent findings from CT can be extrapolated to other places?

Response: Thank you for the suggestion. We have included this sentence in the discussion in this regard: “Our findings may be pertinent to other state prison systems as the national data reveal that cancer is now the leading cause of death among incarcerated individuals.”

Location: Page 18, lines 318-320

6. Review Comments to the Author – Reviewer 2

Methods

Comment: Page 6, line 101: Incarceration data was from 2005-2016. Was this linked to cancer registry data from the same time period?

Response: Yes, the incarceration data from the Connecticut Department of Correction was linked with the Connecticut Tumor Registry from the same period. The linkage was done using name, date of birth, sex, race, ethnicity, and social security number. This is also described in the methods section. 

 Location: We have clarified this in the study design section by adding “from the same period” to text can be seen on page 6, lines 105

Comment: Page 7, lines 127-131: How were patients categorized if they were previously incarcerated > 1 year from their cancer diagnosis?

Response: Patients were categorized based on the timing of their incarceration status in relation to their cancer diagnosis. The three categories are: never incarcerated, currently incarcerated, or recently released from incarceration in the last 12 months. Hence, if a patient was diagnosed with cancer while living in the community, but had been released from incarceration >12 months prior to their cancer diagnosis, we excluded them.

Location: No changes made.

Comment: Page 7, line 141-142. Screenable cancers: Recognizing the explanation provided in the limitations section, consider adding lung and liver cancer to the screenable cancers. Though these screening mechanisms are not part of incarceration screening guidelines (e.g. NCCHC, federal BOP), inclusion of these cancer types then leads to data that helps illustrate the extent to which the current cancer burden can be ameliorated with all available screening modalities.

Response: We considered the inclusion of lung and liver cancer to our analysis of screenable cancers. However, national consensus guidelines for these cancer types have only been recently refined for general population and widely adopted in community practice the last 2-3 years, which is beyond our study period ending in 2016. Because the screening of these cancers was not widely implemented during our study period, we did not classify them as such. However, in our discussion section we have added the following sentence: “We excluded lung and liver cancer from the screenable cancer group because there were no broadly-implemented screening guidelines applicable to these two cancers during our study period.”

Location: Page 8, lines 150-152

Comment: Page 9, lines 166-168. Consider using cancer category as a variable in place of screenable cancers as a variable in the overall survival model. Simplifying cancer types to “screening cancers” may mask significant imbalance of cancers like cervical, breast and colon that have very different prognoses. For instance, never-incarcerated has 16% breast cancer vs 2-5% for the other groups and breast cancer generally has a favorable prognosis compared to many other cancers. Adding at least a sensitivity analysis using cancer category in place of screenable cancer can ensure that cancer mix is not explaining survival differences. This variable could also be included in the survival model for the subset of screenable cancers for the same reason (results presented on page 11, line 206)

If available, a breakdown of cancer diagnosis / survival by incarceration in jail vs prison (if those are separate institutions within the state) would be beneficial.

Response: Thank you for the suggestion, and we agree such breakdown would be insightful. However, we are unable to perform this analysis given the Department of Correction in the state of Connecticut is a unified system (meaning jails and prisons are managed within one system), and even within one facility there are people being held in jail and in prison. Thus, we focused on contact with the Department of Correction, as we can ascertain length of time incarcerated.

Secondly, we were unable to adjust for specific cancer types due to lack of sufficient number of patients for certain types of cancers. Even when the Cox regression converged, the parameter estimates are not stable (they are subject to change if we have a different sample size) and most likely biased. In addition, gender-specific cancers by nature are confounded with sex (also included in the Cox regression as a covariate), leaving cells/blocks mutually exclusive (female reproductive type of cancer will be 0 for male participants). Thus, we decided to aggregate cancer types into larger groups, screenable vs non-screenable, for our final Cox regression models.

Location: No changes made.

Results

Comment: Page 9: Please provide median age of diagnosis.

Response: Thank you for the inquiry. We have updated our results section to include this sentence: “The median age at diagnosis was 50 years for those incarcerated, 51 years for those within 12 months post-release and 66 years for those never incarcerated.” 

Location: Page 9, lines 190-191

Comment: Page 10, line 190: “For screenable cancers, 58.8% were diagnosed at a late stage…”. Please move the definition of “late stage” up from the next sentence. Consider providing this data for distant disease instead of late disease, so it can be compared to the 42.7% data point in the previous sentence. I personally think of “late stage” as representing incurable, distant disease rather than also encompassing locally advanced disease that is still potentially curable. 

Response: With regards to separating late stage and distant, we used late stage and distant stage interchangeably in the paper, hence the confusion about separating locally advanced from metastatic patients. Distant disease is defined as Stage IV cancers, and this definition is stated in the “measures” section. Based on this, we do not have additional analysis to provide as we now have 4 groups of cancer stages: localized, regional, distant and unstaged/unknown. We have edited the manuscript to solely use distant stage as the descriptor for metastatic/stage 4 patients 

Location: Page 10, lines 204,206; page 17, line 303.

Comment: Thinking about how this data can be used to direct implementation of interventions to improve cancer care:

Page 10 line 192-194: Among incarcerated patients with a screenable cancer that were diagnosed with a late stage, the median length of incarceration was 5.9 years, but what percentage of patients had been incarcerated for 1 year or longer? It is another way of illustrating what proportion of cancers might be caught earlier.

Response: Thank you for the feedback and insight on future intervention/policy initiative. We have included this sentence in our results section: “In the incarcerated group 72.4% had been incarcerated for a year or longer and 26.1% in recently released group were incarcerated for at least 1 year.” We have also included a sentence in our discussion about how this data can be used: “In addition, because the majority of those diagnosed with cancer while incarcerated had been incarcerated for more than a year, targeted cancer screening for those incarcerated for more than a year should be implemented.”

Location: Pages 9-10, line 191-193; pages 17-18, lines 316-318

Comment: Among incarcerated and formerly incarcerated patients with screenable cancers diagnosed at a late stage, what is the breakdown of cancer type? This determines which screen programs need to be beefed up.

Response: We thank the reviewer for the suggestion. However, as indicated in our response to reviewer one above, we are unable to provide a detailed breakdown by cancer type among screenable cancers. We grouped them given the small number of individual cases so that we would limit the likelihood of unintentionally revealing the identity of participants.

Location: No changes made.

Comment: The authors mention Medicaid expansion in the discussion – if the registry reports insurance status at diagnosis for recently incarcerated and the general population that would be interesting to see.

Response: Thank you for this suggestion. Unfortunately, we do not have insurance status data to answer this question. Of note, the state of Connecticut expanded Medicaid to include justice involve individuals prior to the implementation of the Affordable Care Act so many of those released likely had access to Medicaid upon release.

Location: No changes made.

---

## [Editor Report · Decision Letter 1]

2 Sep 2022

Incarceration status and cancer mortality: A population-based study

PONE-D-22-07747R1

Dear Dr. Aminawung,

We’re pleased to inform you that your manuscript has been judged scientifically suitable for publication and will be formally accepted for publication once it meets all outstanding technical requirements.

Kind regards,

Andrea Knittel

Academic Editor

PLOS ONE
---

## [Editor Report · Acceptance letter]

8 Sep 2022

PONE-D-22-07747R1 

Incarceration status and cancer mortality: A population-based study 

Dear Dr. Oladeru:

I'm pleased to inform you that your manuscript has been deemed suitable for publication in PLOS ONE. Congratulations! Your manuscript is now with our production department. 

Kind regards, 

on behalf of

Dr. Andrea Knittel 

Academic Editor

PLOS ONE